# Finite-Sample Analysis of Fixed-$k$ Nearest Neighbor Density Functional Estimators

**Shashank Singh**
Statistics & Machine Learning Departments
Carnegie Mellon University
sss1@andrew.cmu.edu

**Barnabás Póczos**
Machine Learning Departments
Carnegie Mellon University
bapoczos@cs.cmu.edu

## Abstract

We provide finite-sample analysis of a general framework for using $k$-nearest neighbor statistics to estimate functionals of a nonparametric continuous probability density, including entropies and divergences. Rather than plugging a consistent density estimate (which requires $k \to \infty$ as the sample size $n \to \infty$) into the functional of interest, the estimators we consider fix $k$ and perform a bias correction. This is more efficient computationally, and, as we show in certain cases, statistically, leading to faster convergence rates. Our framework unifies several previous estimators, for most of which ours are the first finite sample guarantees.

## 1   Introduction

Estimating entropies and divergences of probability distributions in a consistent manner is of importance in a number of problems in machine learning. Entropy estimators have applications in goodness-of-fit testing [13], parameter estimation in semi-parametric models [51], studying fractal random walks [3], and texture classification [14, 15]. Divergence estimators have been used to generalize machine learning algorithms for regression, classification, and clustering from inputs in $\mathbb{R}^D$ to sets and distributions [40, 33].

Divergences also include mutual informations as a special case; mutual information estimators have applications in feature selection [35], clustering [2], causality detection [16], optimal experimental design [26, 38], fMRI data analysis [7], prediction of protein structures [1], and boosting and facial expression recognition [41]. Both entropy estimators and mutual information estimators have been used for independent component and subspace analysis [23, 47, 37, 17], as well as for image registration [14, 15]. Further applications can be found in [25].

This paper considers the more general problem of estimating functionals of the form

$$F(P) := \mathop{\mathbb{E}}_{X \sim P} [f(p(X))], \tag{1}$$

using $n$ IID samples from $P$, where $P$ is an unknown probability measure with smooth density function $p$ and $f$ is a known smooth function. We are interested in analyzing a class of nonparametric estimators based on $k$-nearest neighbor ($k$-NN) distance statistics. Rather than plugging a consistent estimator of $p$ into (1), which requires $k \to \infty$ as $n \to \infty$, these estimators derive a bias correction for the plug-in estimator with *fixed* $k$; hence, we refer to this type of estimator as a fixed-$k$ estimator. Compared to plug-in estimators, fixed-$k$ estimators are faster to compute. As we show, fixed-$k$ estimators can also exhibit superior rates of convergence.

As shown in Table 1, several authors have derived bias corrections necessary for fixed-$k$ estimators of entropies and divergences, including, most famously, the Shannon entropy estimator of [20]. [1] The estimators in Table 1 estimators are known to be weakly consistent, [2] but, except for Shannon entropy,

| Functional Name | Functional Form | Bias Correction | Ref. |
|---|---|---|---|
| Shannon Entropy | $\mathbb{E}\left[\log p(X)\right]$ | Additive constant: $\psi(n) - \psi(k) + \log(k/n)$ | [20][13] |
| Rényi-$\alpha$ Entropy | $\mathbb{E}\left[p^{\alpha-1}(X)\right]$ | Multiplicative constant: $\frac{\Gamma(k)}{\Gamma(k+1-\alpha)}$ | [25, 24] |
| KL Divergence | $\mathbb{E}\left[\log \frac{p(X)}{q(X)}\right]$ | None* | [50] |
| $\alpha$-Divergence | $\mathbb{E}\left[\left(\frac{p(X)}{q(X)}\right)^{\alpha-1}\right]$ | Multiplicative constant: $\frac{\Gamma^2(k)}{\Gamma(k-\alpha+1)\Gamma(k+\alpha-1)}$ | [39] |

Table 1: Functionals with known bias-corrected $k$-NN estimators, their bias corrections, and references. All expectations are over $X \sim P$. $\Gamma(t) = \int_0^\infty x^{t-1} e^{-x}\, dx$ is the gamma function, and $\psi(x) = \frac{d}{dx}\log\left(\Gamma(x)\right)$ is the digamma function. $\alpha \in \mathbb{R}\backslash\{1\}$ is a free parameter. *For KL divergence, bias corrections for $p$ and $q$ cancel.

no finite sample bounds are known. The **main goal of this paper** is to provide finite-sample analysis of these estimators, via unified analysis of the estimator after bias correction. Specifically, we show conditions under which, for $\beta$-Hölder continuous ($\beta \in (0, 2]$) densities on $D$ dimensional space, the bias of fixed-$k$ estimators decays as $O\left(n^{-\beta/D}\right)$ and the variance decays as $O\left(n^{-1}\right)$, giving a mean squared error of $O\left(n^{-2\beta/D} + n^{-1}\right)$. Hence, the estimators converge at the parametric $O(n^{-1})$ rate when $\beta \geq D/2$, and at the slower rate $O(n^{-2\beta/D})$ otherwise. A modification of the estimators would be necessary to leverage additional smoothness for $\beta > 2$, but we do not pursue this here. Along the way, we prove a finite-sample version of the useful fact [25] that (normalized) $k$-NN distances have an Erlang asymptotic distribution, which may be of independent interest.

We present our results for distributions $P$ supported on the unit cube in $\mathbb{R}^D$ because this significantly simplifies the statements of our results, but, as we discuss in the supplement, our results generalize fairly naturally, for example to distributions supported on smooth compact manifolds. In this context, it is worth noting that our results scale with the *intrinsic* dimension of the manifold. As we discuss later, we believe deriving finite sample rates for distributions with *unbounded* support may require a truncated modification of the estimators we study (as in [49]), but we do not pursue this here.

## 2 Problem statement and notation

Let $\mathcal{X} := [0, 1]^D$ denote the unit cube in $\mathbb{R}^D$, and let $\mu$ denote the Lebesgue measure. Suppose $P$ is an unknown $\mu$-absolutely continuous Borel probability measure supported on $\mathcal{X}$, and let $p : \mathcal{X} \to [0, \infty)$ denote the density of $P$. Consider a (known) differentiable function $f : (0, \infty) \to \mathbb{R}$. Given $n$ samples $X_1, ..., X_n$ drawn IID from $P$, we are interested in estimating the functional

$$F(P) := \mathbb{E}_{X \sim P}\left[f(p(X))\right].$$

Somewhat more generally (as in divergence estimation), we may have a function $f : (0, \infty)^2 \to \mathbb{R}$ of two variables and a second unknown probability measure $Q$, with density $q$ and $n$ IID samples $Y_1, ..., Y_n$. Then, we are interested in estimating

$$F(P, Q) := \mathbb{E}_{X \sim P}\left[f(p(X), q(X))\right].$$

Fix $r \in [1, \infty]$ and a positive integer $k$. We will work with distances induced by the $r$-norm

$$\|x\|_r := \left(\sum_{i=1}^D x_i^r\right)^{1/r} \quad \text{and define} \quad c_{D,r} := \frac{(2\Gamma(1 + 1/r))^D}{\Gamma(1 + D/r)} = \mu(B(0, 1)),$$

where $B(x, \varepsilon) := \{y \in \mathbb{R}^D : \|x - y\|_r < \varepsilon\}$ denotes the open radius-$\varepsilon$ ball centered at $x$. Our estimators use $k$-nearest neighbor ($k$-NN) distances:

**Definition 1. ($k$-NN distance):** *Given $n$ IID samples $X_1, ..., X_n$ from $P$, for $x \in \mathbb{R}^D$, we define the $k$-NN distance $\varepsilon_k(x)$ by $\varepsilon_k(x) = \|x - X_i\|_r$, where $X_i$ is the $k^{th}$-nearest element (in $\|\cdot\|_r$) of the set $\{X_1, ..., X_n\}$ to $x$. For divergence estimation, given $n$ samples $Y_1, ..., Y_n$ from $Q$, then we similarly define $\delta_k(x)$ by $\delta_k(x) = \|x - Y_i\|_r$, where $Y_i$ is the $k^{th}$-nearest element of $\{Y_1, ..., Y_n\}$ to $x$.*

$\mu$-absolute continuity of $P$ precludes the existence of atoms (i.e., $\forall x \in \mathbb{R}^D$, $P(\{x\}) = \mu(\{x\}) = 0$). Hence, each $\varepsilon_k(x) > 0$ a.s. We will require this to study quantities such as $\log \varepsilon_k(x)$ and $1/\varepsilon_k(x)$.

## 3  Estimator

### 3.1  $k$-NN density estimation and plug-in functional estimators

The $k$-NN density estimator

$$\hat{p}_k(x) = \frac{k/n}{\mu(B(x, \varepsilon_k(x)))} = \frac{k/n}{c_D \varepsilon_k^D(x)}$$

is well-studied nonparametric density estimator [28], motivated by noting that, for small $\varepsilon > 0$,

$$p(x) \approx \frac{P(B(x, \varepsilon))}{\mu(B(x, \varepsilon))},$$

and that, $P(B(x, \varepsilon_k(x))) \approx k/n$. One can show that, for $x \in \mathbb{R}^D$ at which $p$ is continuous, if $k \to \infty$ and $k/n \to 0$ as $n \to \infty$, then $\hat{p}_k(x) \to p(x)$ in probability ([28], Theorem 3.1). Thus, a natural approach for estimating $F(P)$ is the plug-in estimator

$$\hat{F}_{PI} := \frac{1}{n} \sum_{i=1}^{n} f\left(\hat{p}_k(X_i)\right). \tag{2}$$

Since $\hat{p}_k \to p$ in probability pointwise as $k, n \to \infty$ and $f$ is smooth, one can show $\hat{F}_{PI}$ is consistent, and in fact derive finite sample convergence rates (depending on how $k \to \infty$). For example, [44] show a convergence rate of $O\left(n^{-\min\left\{\frac{2\beta}{\beta+D}, 1\right\}}\right)$ for $\beta$-Hölder continuous densities (after sample splitting and boundary correction) by setting $k \asymp n^{\frac{\beta}{\beta+d}}$.

Unfortunately, while necessary to ensure $\mathbb{V}\left[\hat{p}_k(x)\right] \to 0$, the requirement $k \to \infty$ is computationally burdensome. Furthermore, increasing $k$ can increase the bias of $\hat{p}_k$ due to over-smoothing (see (5) below), suggesting that this may be sub-optimal for estimating $F(P)$. Indeed, similar work based on kernel density estimation [42] suggests that, for plug-in functional estimators, *under-smoothing* may be preferable, since the empirical mean results in additional smoothing.

### 3.2  Fixed-$k$ functional estimators

An alternative approach is to fix $k$ as $n \to \infty$. Since $\hat{F}_{PI}$ is itself an empirical mean, unlike $\mathbb{V}\left[\hat{p}_k(x)\right]$, $\mathbb{V}\left[\hat{F}_{PI}\right] \to 0$ as $n \to \infty$. The more critical complication of fixing $k$ is bias. Since $f$ is typically non-linear, the non-vanishing variance of $\hat{p}_k$ translates into asymptotic bias. A solution adopted by several papers is to derive a bias correction function $\mathcal{B}$ (depending only on known factors) such that

$$\mathop{\mathbb{E}}_{X_1, \ldots, X_n}\left[\mathcal{B}\left(f\left(\frac{k/n}{\mu(B(x, \varepsilon_k(x)))}\right)\right)\right] = \mathop{\mathbb{E}}_{X_1, \ldots, X_n}\left[f\left(\frac{P(B(x, \varepsilon_k(x)))}{\mu(B(x, \varepsilon_k(x)))}\right)\right]. \tag{3}$$

For continuous $p$, the quantity

$$p_{\varepsilon_k(x)}(x) := \frac{P(B(x, \varepsilon_k(x)))}{\mu(B(x, \varepsilon_k(x)))} \tag{4}$$

*is* a consistent estimate of $p(x)$ with $k$ fixed, but it is not computable, since $P$ is unknown. The bias correction $\mathcal{B}$ gives us an asymptotically unbiased estimator

$$\hat{F}_{\mathcal{B}}(P) := \frac{1}{n} \sum_{i=1}^{n} \mathcal{B}\left(f\left(\hat{p}_k(X_i)\right)\right) = \frac{1}{n} \sum_{i=1}^{n} \mathcal{B}\left(f\left(\frac{k/n}{\mu(B(X_i, \varepsilon_k(X_i)))}\right)\right).$$

that uses $k/n$ in place of $P(B(x, \varepsilon_k(x)))$. This estimate extends naturally to divergences:

$$\hat{F}_{\mathcal{B}}(P, Q) := \frac{1}{n} \sum_{i=1}^{n} \mathcal{B}\left(f\left(\hat{p}_k(X_i), \hat{q}_k(X_i)\right)\right).$$

As an example, if $f = \log$ (as in Shannon entropy), then it can be shown that, for any continuous $p$,

$$\mathbb{E}\left[\log P(B(x, \varepsilon_k(x)))\right] = \psi(k) - \psi(n).$$

Hence, for $B_{n,k} := \psi(k) - \psi(n) + \log(n) - \log(k)$,

$$\mathop{\mathbb{E}}_{X_1,\ldots,X_n}\left[f\left(\frac{k/n}{\mu(B(x,\varepsilon_k(x)))}\right)\right] + B_{n,k} = \mathop{\mathbb{E}}_{X_1,\ldots,X_n}\left[f\left(\frac{P(B(x,\varepsilon_k(x)))}{\mu(B(x,\varepsilon_k(x)))}\right)\right].$$

giving the estimator of [20]. Other examples of functionals for which the bias correction is known are given in Table 1.

In general, deriving an appropriate bias correction can be quite a difficult problem specific to the functional of interest, and it is not our goal presently to study this problem; rather, we are interested in bounding the error of $\hat{F}_{\mathcal{B}}(P)$, *assuming the bias correction is known.* Hence, our results apply to all of the estimators in Table 1, as well as any estimators of this form that may be derived in the future.

## 4  Related work

### 4.1  Estimating information theoretic functionals

Recently, there has been much work on analyzing estimators for entropy, mutual information, divergences, and other functionals of densities. Besides bias-corrected fixed-$k$ estimators, most of this work has taken one of three approaches. One series of papers [27, 42, 43] studied a boundary-corrected plug-in approach based on under-smoothed kernel density estimation. This approach has strong finite sample guarantees, but requires prior knowledge of the support of the density, and can have a slow rate of convergence. A second approach [18, 22] uses von Mises expansion to partially correct the bias of optimally smoothed density estimates. This is statistically more efficient, but can require computationally demanding numerical integration over the support of the density. A final line of work [30, 31, 44, 46] studied plug-in estimators based on consistent, boundary corrected $k$-NN density estimates (i.e., with $k \to \infty$ as $n \to \infty$). [32] study a divergence estimator based on convex risk minimization, but this relies of the context of an RKHS, making results are difficult to compare.

**Rates of Convergence:** For densities over $\mathbb{R}^D$ satisfying a Hölder smoothness condition parametrized by $\beta \in (0,\infty)$, the minimax mean squared error rate for estimating functionals of the form $\int f(p(x))\,dx$ has been known since [6] to be $O\left(n^{-\min\left\{\frac{8\beta}{4\beta+D},1\right\}}\right)$. [22] recently derived identical minimax rates for divergence estimation.

Most of the above estimators have been shown to converge at the rate $O\left(n^{-\min\left\{\frac{2\beta}{\beta+D},1\right\}}\right)$. Only the von Mises approach [22] is known to achieve the minimax rate for general $\beta$ and $D$, but due to its computational demand ($O(2^D n^3)$), [3] the authors suggest using other statistically less efficient estimators for moderate sample size. Here, we show that, for $\beta \in (0,2]$, bias-corrected fixed-$k$ estimators converge at the relatively fast rate $O\left(n^{-\min\left\{\frac{2\beta}{D},1\right\}}\right)$. For $\beta > 2$, modifications are needed for the estimator to leverage the additional smoothness of the density. Notably, this rate is *adaptive*; that is, it does not require selecting a smoothing parameter depending on the unknown $\beta$; our results (Theorem 5) imply the above rate is achieved for *any* fixed choice of $k$. On the other hand, since no empirical error metric is available for cross-validation, parameter selection is an obstacle for competing estimators.

### 4.2  Prior analysis of fixed-$k$ estimators

As of writing this paper, the only finite-sample results for $\hat{F}_{\mathcal{B}}(P)$ were those of [5] for the Kozachenko-Leonenko (KL) [4] Shannon entropy estimator. [20] Theorem 7.1 of [5] shows that, if the density $p$ has compact support, then the variance of the KL estimator decays as $O(n^{-1})$. They also claim (Theorem 7.2) to bound the bias of the KL estimator by $O(n^{-\beta})$, under the assumptions that $p$ is $\beta$-Hölder continuous ($\beta \in (0,1]$), bounded away from $0$, and supported on the interval $[0,1]$. However, in their proof, [5] neglect to bound the additional bias incurred near the boundaries of $[0,1]$, where the density cannot simultaneously be bounded away from $0$ and continuous. In fact, because the KL estimator does not attempt to correct for boundary bias, it is not clear that the bias should decay as $O(n^{-\beta})$ under these conditions; we require additional conditions at the boundary of $\mathcal{X}$.

Finally, two very recent papers [12, 4] have analyzed the KL estimator. In this case, [12] generalize the results of [5] to $D > 1$, and [4] weaken the regularity and boundary assumptions required by our bias bound, while deriving the same rate of convergence. Moreover, they show that, if $k$ increases with $n$ at the rate $k \asymp \log^5 n$, the KL estimator is asymptotically efficient (i.e., asymptotically normal, with optimal asymptotic variance). As explained in Section 8, together with our results this elucidates the role of $k$ in the KL estimator: fixing $k$ optimizes the convergence rate of the estimator, but increasing $k$ slowly can further improve error by constant factors.

## 5   Discussion of assumptions

The lack of finite-sample results for fixed-$k$ estimators is due to several technical challenges. Here, we discuss some of these challenges, motivating the assumptions we make to overcome them.

First, these estimators are sensitive to regions of low probability (i.e., $p(x)$ small), for two reasons:

1. Many functions $f$ of interest (e.g., $f = \log$ or $f(z) = z^\alpha$, $\alpha < 0$) have singularities at 0.
2. The $k$-NN estimate $\hat{p}_k(x)$ of $p(x)$ is highly biased when $p(x)$ is small. For example, for $p$ $\beta$-Hölder continuous ($\beta \in (0, 2]$), one has ([29], Theorem 2)

$$\text{Bias}(\hat{p}_k(x)) \asymp \left( \frac{k}{np(x)} \right)^{\beta/D}. \tag{5}$$

For these reasons, it is common in analysis of $k$-NN estimators to assume the following [5, 39]:

**(A1)** $p$ is bounded away from zero on its support. That is, $p_* := \inf_{x \in \mathcal{X}} p(x) > 0$.

Second, unlike many functional estimators (see e.g., [34, 45, 42]), the fixed-$k$ estimators we consider do not attempt correct for boundary bias (i.e., bias incurred due to discontinuity of $p$ on the boundary $\partial \mathcal{X}$ of $\mathcal{X}$). [5] The boundary bias of the density estimate $\hat{p}_k(x)$ does vanish at $x$ in the interior $\mathcal{X}^\circ$ of $\mathcal{X}$ as $n \to \infty$, but additional assumptions are needed to obtain finite-sample rates. Either of the following assumptions would suffice:

**(A2)** $p$ is continuous not only on $\mathcal{X}^\circ$ but also on $\partial \mathcal{X}$ (i.e., $p(x) \to 0$ as $\text{dist}(x, \partial \mathcal{X}) \to 0$).
**(A3)** $p$ is supported on all of $\mathbb{R}^D$. That is, the support of $p$ has no boundary. This is the approach of [49], but we reiterate that, to handle an unbounded domain, they require truncating $\varepsilon_k(x)$.

Unfortunately, both assumptions **(A2)** and **(A3)** are inconsistent with **(A1)**. Our approach is to assume **(A2)** and replace assumption **(A1)** with a much milder assumption that $p$ is *locally lower bounded* on its support in the following sense:

**(A4)** There exist $\rho > 0$ and a function $p_* : \mathcal{X} \to (0, \infty)$ such that, for all $x \in \mathcal{X}, r \in (0, \rho]$, $p_*(x) \leq \frac{P(B(x,r))}{\mu(B(x,r))}$.

We show in Lemma 2 that assumption **(A4)** is in fact very mild; in a metric measure space of positive dimension $D$, as long as $p$ is continuous on $\mathcal{X}$, such a $p_*$ exists for *any* desired $\rho > 0$. For simplicity, we will use $\rho = \sqrt{D} = \text{diam}(\mathcal{X})$.

As hinted by (5) and the fact that $F(P)$ is an expectation, our bounds will contain terms of the form

$$\mathbb{E}_{X \sim P} \left[ \frac{1}{(p_*(X))^{\beta/D}} \right] = \int_{\mathcal{X}} \frac{p(x)}{(p_*(x))^{\beta/D}} \, d\mu(x)$$

(with an additional $f'(p_*(x))$ factor if $f$ has a singularity at zero). Hence, our key assumption is that these quantities are finite. This depends primarily on *how quickly* $p$ approaches zero near $\partial \mathcal{X}$. For many functionals, Lemma 6 gives a simple sufficient condition.

## 6 Preliminary lemmas

Here, we present some lemmas, both as a means of summarizing our proof techniques and also because they may be of independent interest for proving finite-sample bounds for other $k$-NN methods. Due to space constraints, all proofs are given in the appendix. Our first lemma states that, if $p$ is continuous, then it is locally lower bounded as described in the previous section.

**Lemma 2. (Existence of Local Bounds)** *If $p$ is continuous on $\mathcal{X}$ and strictly positive on the interior $\mathcal{X}^\circ$ of $\mathcal{X}$, then, for $\rho := \sqrt{D} = \mathrm{diam}(\mathcal{X})$, there exists a continuous function $p_* : \mathcal{X}^\circ \to (0, \infty)$ and a constant $p^* \in (0, \infty)$ such that*

$$0 < p_*(x) \leq \frac{P(B(x,r))}{\mu(B(x,r))} \leq p^* < \infty, \quad \forall x \in \mathcal{X}, r \in (0, \rho].$$

We now use these local lower and upper bounds to prove that $k$-NN distances concentrate around a term of order $(k/(np(x)))^{1/D}$. Related lemmas, also based on multiplicative Chernoff bounds, are used by [21, 9] and [8, 19] to prove finite-sample bounds on $k$-NN methods for cluster tree pruning and classification, respectively. For cluster tree pruning, the relevant inequalities bound the error of the $k$-NN density estimate, and, for classification, they lower bound the probability of nearby samples of the same class. Unlike in cluster tree pruning, we are not using a consistent density estimate, and, unlike in classification, our estimator is a function of $k$-NN distances themselves (rather than their ordering). Thus, our statement is somewhat different, bounding the $k$-NN distances themselves:

**Lemma 3. (Concentration of $k$-NN Distances)** *Suppose $p$ is continuous on $\mathcal{X}$ and strictly positive on $\mathcal{X}^\circ$. Let $p_*$ and $p^*$ be as in Lemma 2. Then, for any $x \in \mathcal{X}^\circ$,*

*1. if $r > \left(\frac{k}{p_*(x)n}\right)^{1/D}$, then $\mathbb{P}\left[\varepsilon_k(x) > r\right] \leq e^{-p_*(x)r^D n}\left(e\frac{p_*(x)r^D n}{k}\right)^k$.*

*2. if $r \in \left[0, \left(\frac{k}{p^* n}\right)^{1/D}\right)$, then $\mathbb{P}\left[\varepsilon_k(x) < r\right] \leq e^{-p_*(x)r^D n}\left(\frac{ep^* r^D n}{k}\right)^{kp_*(x)/p^*}$.*

It is worth noting an asymmetry in the above bounds: counter-intuitively, the lower bound depends on $p_*$. This asymmetry is related to the large bias of $k$-NN density estimators when $p$ is small (as in (5)).

The next lemma uses Lemma 3 to bound expectations of monotone functions of the ratio $\hat{p}_k/p_*$. As suggested by the form of integrals (6) and (7), this is essentially a finite-sample statement of the fact that (appropriately normalized) $k$-NN distances have Erlang asymptotic distributions; this asymptotic statement is key to consistency proofs of [25] and [39] for $\alpha$-entropy and divergence estimators.

**Lemma 4.** *Let $p$ be continuous on $\mathcal{X}$ and strictly positive on $\mathcal{X}^\circ$. Define $p_*$ and $p^*$ as in Lemma 2. Suppose $f : (0, \infty) \to \mathbb{R}$ is continuously differentiable and $f' > 0$. Then, we have the upper bound* [6]

$$\sup_{x \in \mathcal{X}^\circ} \mathbb{E}\left[f_+\left(\frac{p_*(x)}{\hat{p}_k(x)}\right)\right] \leq f_+(1) + e\sqrt{k} \int_k^\infty \frac{e^{-y}y^k}{\Gamma(k+1)} f_+\left(\frac{y}{k}\right) dy, \tag{6}$$

*and, for all $x \in \mathcal{X}^\circ$, for $\kappa(x) := kp_*(x)/p^*$, the lower bound*

$$\mathbb{E}\left[f_-\left(\frac{p_*(x)}{\hat{p}_k(x)}\right)\right] \leq f_-(1) + e\sqrt{\frac{k}{\kappa(x)}} \int_0^{\kappa(x)} \frac{e^{-y}y^{\kappa(x)}}{\Gamma(\kappa(x)+1)} f_-\left(\frac{y}{k}\right) dy \tag{7}$$

Note that plugging the function $z \mapsto f\left(\left(\frac{kz}{c_{D,r}np_*(x)}\right)^{\frac{1}{D}}\right)$ into Lemma 4 gives bounds on $\mathbb{E}\left[f(\varepsilon_k(x))\right]$. As one might guess from Lemma 3 and the assumption that $f$ is smooth, this bound is roughly of the order $\asymp \left(\frac{k}{np(x)}\right)^{\frac{1}{D}}$. For example, for any $\alpha > 0$, a simple calculation from (6) gives

$$\mathbb{E}\left[\varepsilon_k^\alpha(x)\right] \leq \left(1 + \frac{\alpha}{D}\right)\left(\frac{k}{c_{D,r}np_*(x)}\right)^{\frac{\alpha}{D}}. \tag{8}$$

(8) is used for our bias bound, and more direct applications of Lemma 4 are used in variance bound.

# 7 Main results

Here, we present our main results on the bias and variance of $\hat{F}_{\mathcal{B}}(P)$. Again, due to space constraints, all proofs are given in the appendix. We begin with bounding the bias:

**Theorem 5. (Bias Bound)** *Suppose that, for some $\beta \in (0, 2]$, $p$ is $\beta$-Hölder continuous with constant $L > 0$ on $\mathcal{X}$, and $p$ is strictly positive on $\mathcal{X}^\circ$. Let $p_*$ and $p^*$ be as in Lemma 2. Let $f : (0, \infty) \to \mathbb{R}$ be differentiable, and define $M_{f,p} : \mathcal{X} \to [0, \infty)$ by*

$$M_{f,p}(x) := \sup_{z \in [p_*(x), p^*]} \left| \frac{d}{dz} f(z) \right|$$

*Assume*

$$C_f := \mathop{\mathbb{E}}_{X \sim p} \left[ \frac{M_{f,p}(X)}{(p_*(X))^{\frac{\beta}{D}}} \right] < \infty. \quad \textit{Then,} \quad \left| \mathbb{E}\, \hat{F}_{\mathcal{B}}(P) - F(P) \right| \leq C_f L \left( \frac{k}{n} \right)^{\frac{\beta}{D}}.$$

The statement for divergences is similar, assuming that $q$ is also $\beta$-Hölder continuous with constant $L$ and strictly positive on $\mathcal{X}^\circ$. Specifically, we get the same bound if we replace $M_{f,o}$ with

$$M_{f,p}(x) := \sup_{(w,z) \in [p_*(x), p^*] \times [q_*(x), q^*]} \left| \frac{\partial}{\partial w} f(w, z) \right|$$

and define $M_{f,q}$ similarly (i.e., with $\frac{\partial}{\partial z}$) and we assume that

$$C_f := \mathop{\mathbb{E}}_{X \sim p} \left[ \frac{M_{f,p}(X)}{(p_*(X))^{\frac{\beta}{D}}} \right] + \mathop{\mathbb{E}}_{X \sim p} \left[ \frac{M_{f,q}(X)}{(q_*(X))^{\frac{\beta}{D}}} \right] < \infty.$$

As an example of the applicability of Theorem 5, consider estimating the Shannon entropy. Then, $f(z) = \log(x)$, and so we need $C_f = \int_{\mathcal{X}} (p_*(x))^{-\beta/D} \, d\mu(x) < \infty$.

The assumption $C_f < \infty$ is not immediately transparent. For the functionals in Table 1, $C_f$ has the form $\int_{\mathcal{X}} (p(x))^{-c} \, dx$, for some $c > 0$, and hence $C_f < \infty$ intuitively means $p(x)$ cannot approach zero too quickly as $\text{dist}(x, \partial \mathcal{X}) \to 0$. The following lemma gives a formal sufficient condition:

**Lemma 6. (Boundary Condition)** *Let $c > 0$. Suppose there exist $b_\partial \in (0, \frac{1}{c})$, $c_\partial, \rho_\partial > 0$ such that, for all $x \in \mathcal{X}$ with $\varepsilon(x) := \text{dist}(x, \partial \mathcal{X}) < \rho_\partial$, $p(x) \geq c_\partial \varepsilon^{b_\partial}(x)$. Then, $\int_{\mathcal{X}} (p_*(x))^{-c} \, d\mu(x) < \infty$.*

In the supplement, we give examples showing that this condition is fairly general, satisfied by densities proportional to $x^{b_\partial}$ near $\partial \mathcal{X}$ (i.e., those with at least $b_\partial$ nonzero one-sided derivatives on the boundary).

We now bound the variance. The main obstacle here is that the fixed-$k$ estimator is an empirical mean of *dependent* terms (functions of $k$-NN distances). We generalize the approach used by [5] to bound the variance of the KL estimator of Shannon entropy. The key insight is the geometric fact that, in $(\mathbb{R}^D, \|\cdot\|_p)$, there exists a constant $N_{k,D}$ (independent of $n$) such that any sample $X_i$ can be amongst the $k$-nearest neighbors of at most $N_{k,D}$ other samples. Hence, at most $N_{k,D} + 1$ of the terms in (2) can change when a single $X_i$ is added, suggesting a variance bound via the Efron-Stein inequality [10], which bounds the variance of a function of random variables in terms of its expected change when its arguments are resampled. [11] originally used this approach to prove a general Law of Large Numbers (LLN) for nearest-neighbors statistics. Unfortunately, this LLN relies on bounded kurtosis assumptions that are difficult to justify for the $\log$ or negative power statistics we study.

**Theorem 7. (Variance Bound)** *Suppose $\mathcal{B} \circ f$ is continuously differentiable and strictly monotone. Assume $C_{f,p} := \mathbb{E}_{X \sim P} \left[ \mathcal{B}^2(f(p_*(X))) \right] < \infty$, and $C_f := \int_0^\infty e^{-y} y^k f(y) < \infty$. Then, for*

$$C_V := 2 \left( 1 + N_{k,D} \right) \left( 3 + 4k \right) \left( C_{f,p} + C_f \right), \quad \textit{we have} \quad \mathbb{V}\left[ \hat{F}_{\mathcal{B}}(P) \right] \leq \frac{C_V}{n}.$$

As an example, if $f = \log$ (as in Shannon entropy), then, since $\mathcal{B}$ is an additive constant, we simply require $\int_{\mathcal{X}} p(x) \log^2(p_*(x)) < \infty$. In general, $N_{k,D}$ is of the order $k2^{cD}$, for some $c > 0$. Our bound is likely quite loose in $k$; in practice, $\mathbb{V}\left[ \hat{F}_{\mathcal{B}}(P) \right]$ typically decreases somewhat with $k$.

# 8   Conclusions and discussion

In this paper, we gave finite-sample bias and variance error bounds for a class of fixed-$k$ estimators of functionals of probability density functions, including the entropy and divergence estimators in Table 1. The bias and variance bounds in turn imply a bound on the mean squared error (MSE) of the bias-corrected estimator via the usual decomposition into squared bias and variance:

**Corollary 8.  (MSE Bound)** *Under the conditions of Theorems 5 and 7,*

$$\mathbb{E}\left[\left(\hat{H}_k(X) - H(X)\right)^2\right] \leq C_f^2 L^2 \left(\frac{k}{n}\right)^{2\beta/D} + \frac{C_V}{n}. \tag{9}$$

**Choosing $k$:** Contrary to the name, fixing $k$ is not *required* for "fixed-$k$" estimators. [36] empirically studied the effect of changing $k$ with $n$ and found that fixing $k = 1$ gave best results for estimating $F(P)$. However, there has been no theoretical justification for fixing $k$. Assuming tightness of our bias bound in $k$, we provide this in a worst-case sense: since our bias bound is nondecreasing in $k$ and our variance bound is no larger than the minimax MSE rate for these estimation problems, reducing variance (i.e., increasing $k$) does not improve the (worst-case) convergence rate. On the other hand, [4] recently showed that slowly increasing $k$ can improves the asymptotic variance of the estimator, with the rate $k \asymp \log^5 n$ leading to asymptotic efficiency. In view of these results, we suggest that increasing $k$ can improve error by constant factors, but cannot improve the convergence rate.

Finally, we note that [36] found increasing $k$ quickly (e.g., $k = n/2$) was *best* for certain hypothesis tests based on these estimators. Intuitively, this is because, in testing problems, bias is less problematic than variance (e.g., an asymptotically biased estimator can still lead to a consistent test).

### Acknowledgments

This material is based upon work supported by a National Science Foundation Graduate Research Fellowship to the first author under Grant No. DGE-1252522.

## Footnotes

[1]MATLAB code for these estimators is in the ITE toolbox https://bitbucket.org/szzoli/ite/ [48].

[2]Several of these proofs contain errors regarding the use of integral convergence theorems when their conditions do not hold, as described in [39].

[3]Fixed-$k$ estimators can be computed in $O\left(Dn^2\right)$ time, or $O\left(2^D n \log n\right)$ using $k$-d trees for small $D$.

[4]Not to be confused with Kullback-Leibler (KL) divergence, for which we also analyze an estimator.

[49] studied a closely related entropy estimator for which they prove $\sqrt{n}$-consistency. Their estimator is identical to the KL estimator, except that it truncates $k$-NN distances at $\sqrt{n}$, replacing $\varepsilon_k(x)$ with $\min\{\varepsilon_k(x), \sqrt{n}\}$. This sort of truncation may be necessary for certain fixed-$k$ estimators to satisfy finite-sample bounds for densities of *unbounded* support, though consistency can be shown regardless.

[5] This complication was omitted in the bias bound (Theorem 7.2) of [5] for entropy estimation.

[6] $f_+(x) = \max\{0, f(x)\}$ and $f_-(x) = -\min\{0, f(x)\}$ denote the positive and negative parts of $f$. Recall that $\mathbb{E}\left[f(X)\right] = \mathbb{E}\left[f_+(X)\right] - \mathbb{E}\left[f_-(X)\right]$.

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
