[Supplementary Material · nips_2016_supplement.pdf]

# Finite-Sample Analysis of Fixed-$k$ Nearest Neighbor Density Functional Estimators

**Shashank Singh**
Statistics & Machine Learning Departments
Carnegie Mellon University
sss1@andrew.cmu.edu

**Barnabás Póczos**
Machine Learning Departments
Carnegie Mellon University
bapoczos@cs.cmu.edu

## A    A More General Setting

In the main paper, for the sake of clarity, we discussed only the setting of distributions on the $D$-dimensional unit cube $[0, 1]^D$. For sake of generality, we prove our results in the significantly more general setting of a set equipped with a metric, a base measure, a probability density, and an appropriate definition of dimension. This setting subsumes Euclidean spaces, in which $k$-NN methods are usually analyzed, but also includes, for instance, Riemannian manifolds.

**Definition 1. (Metric Measure Space):** *A quadruple $(\mathbb{X}, d, \Sigma, \mu)$ is called a* metric measure space *if $(\mathbb{X}, d)$ is a complete metric space, $(\mathbb{X}, \Sigma, \mu)$ is a $\sigma$-finite measure space, and $\Sigma$ contains the Borel $\sigma$-algebra induced by $d$.*

**Definition 2. (Scaling Dimension):** *A metric measure space $(\mathbb{X}, d, \Sigma, \mu)$ has* scaling dimension *$D \in [0, \infty)$ if there exist constants $\mu_*, \mu^* > 0$ such that, $\forall r > 0$, $x \in \mathbb{X}$, $\mu_* \leq \frac{\mu(B(x,r))}{r^D} \leq \mu^*$. [1]*

**Remark 3.** *The above definition of dimension coincides with $D$ in $\mathbb{R}^D$, where, under the $L^p$ metric and Lebesgue measure,*

$$\mu_* = \mu^* = \frac{(2\Gamma(1 + 1/p))^D}{\Gamma(1 + D/p)}$$

*is the usual volume of the unit ball. However, it is considerably more general than the vector-space definition of dimension. It includes, for example, the case that $\mathbb{X}$ is a smooth Riemannian manifold, with the standard metric and measure induced by the Riemann metric. In this case, our results scale with the* intrinsic *dimension of data, rather than the dimension of a space in which the data are embedded. Often, $\mu_* = \mu^*$, but leaving these distinct allows, for example, manifolds with boundary. The scaling dimension is slightly more restrictive than the well-studied doubling dimension of a measure, [3] which enforces only an upper bound on the rate of growth.*

## B    Proofs of Lemmas

**Lemma 2.** *Consider a metric measure space $(\mathbb{X}, d, \Sigma, \mu)$ of scaling dimension $D$, and a $\mu$-absolutely continuous probability measure $P$, with density function $p : \mathbb{X} \to [0, \infty)$ supported on*

$$\mathcal{X} := \{x \in \mathbb{X} : p(x) > 0\}.$$

*If $p$ is continuous on $\mathcal{X}$, then, for any $\rho > 0$, there exists a function $p_* : \mathcal{X} \to (0, \infty)$ such that*

$$0 < p_*(x) \leq \inf_{r \in (0, \rho]} \frac{P(B(x, r))}{\mu(B(x, r))}, \quad \forall x \in \mathcal{X},$$

*and, if $p$ is bounded above by $p^* := \sup_{x \in \mathcal{X}} p(x) < \infty$, then*

$$\sup_{r \in (0, \rho]} \frac{P(B(x, r))}{\mu(B(x, r))} \leq p^* < \infty, \quad \forall r \in (0, \rho],$$

**Proof:** Let $x \in \mathcal{X}$. Since $p$ is continuous and strictly positive at $x$, there exists $\varepsilon \in (0, \rho]$ such that and, for all $y \in B(x, \varepsilon)$, $p(y) \geq p(x)/2 > 0$. Define

$$p_*(x) := \frac{p(x)}{2} \frac{\mu_*}{\mu^*} \left(\frac{\varepsilon}{\rho}\right)^D .$$

Then, for any $r \in (0, \rho]$, since $P$ is a non-negative measure, and $\mu$ has scaling dimension $D$,

$$P(B(x, r)) \geq P(B(x, \varepsilon r/\rho)) \geq \mu(B(x, \varepsilon r/\rho)) \min_{y \in B(x, \varepsilon r/\rho)} p(y)$$

$$\geq \mu(B(x, \varepsilon r/\rho)) \frac{p(x)}{2}$$

$$\geq \frac{p(x)}{2} \mu_* \left(\frac{\varepsilon r}{\rho}\right)^D = p_*(x) \mu^* r^D \geq p_*(x) \mu(B(x, r)).$$

Also, trivially, $\forall r \in (0, \rho]$,

$$P(B(x, r)) \leq \mu(B(x, r)) \max_{y \in B(x, r\rho/\varepsilon)} p(y) \leq p^*(x) \mu(B(x, r)).$$

■

**Lemma 3.** *Consider a metric measure space $(\mathbb{X}, d, \Sigma, \mu)$ of scaling dimension $D$, and a $\mu$-absolutely continuous probability measure $P$, with continuous density function $p : \mathbb{X} \to [0, \infty)$ supported on*

$$\mathcal{X} := \{x \in \mathbb{X} : p(x) > 0\}.$$

*For $x \in \mathcal{X}$, if $r > \left(\frac{k}{p_*(x)n}\right)^{1/D}$, then*

$$\mathbb{P}\left[\varepsilon_k(x) > r\right] \leq e^{-p_*(x)r^D n} \left(e\frac{p_*(x)r^D n}{k}\right)^k .$$

*and, if $r \in \left[0, \left(\frac{k}{p^* n}\right)^{1/D}\right)$, then*

$$\mathbb{P}\left[\varepsilon_k(x) \leq r\right] \leq e^{-p_*(x)r^D n} \left(\frac{ep^* r^D n}{k}\right)^{kp_*(x)/p^*} .$$

**Proof:** Notice that, for all $x \in \mathcal{X}$ and $r > 0$,

$$\sum_{i=1}^{n} 1_{\{X_i \in B(x, r)\}} \sim \text{Binomial}\left(n, P(B(x, r))\right),$$

and hence that many standard concentration inequalities apply. Since we are interested in small $r$ (and hence small $P(B(x, r))$), we prefer bounds on relative error, and hence apply multiplicative Chernoff bounds. If $r > (k/(p_*(x)n))^{1/D}$, then, by definition of $p_*$, $P(B(x, r)) < k/n$, and so, applying the multiplicative Chernoff bound with $\delta := \frac{p_*(x)r^D n - k}{p_*(x)r^D n} > 0$ gives

$$\mathbb{P}\left[\varepsilon_k(x) > r\right] = \mathbb{P}\left[\sum_{i=1}^{n} 1_{\{X_i \in B(x, r)\}} < k\right]$$

$$\leq \mathbb{P}\left[\sum_{i=1}^{n} 1_{\{X_i \in B(x, r)\}} < (1 - \delta)nP(B(x, r))\right]$$

$$\leq \left(\frac{e^{-\delta}}{(1 - \delta)^{(1-\delta)}}\right)^{nP(B(x, r))}$$

$$= e^{-p_*(x)r^D n} \left(\frac{ep_*(x)r^D n}{k}\right)^k .$$

Similarly, if $r < (k/(p^*n))^{1/D}$, then, applying the multiplicative Chernoff bound with $\delta := \frac{k - p^* r^D n}{p^* r^D n} > 0$,

$$\mathbb{P}\left[\varepsilon_k(x) < r\right] = \mathbb{P}\left[\sum_{i=1}^n \mathbb{1}_{\{X_i \in B(x,r)\}} \geq k\right]$$

$$\leq \mathbb{P}\left[\sum_{i=1}^n \mathbb{1}_{\{X_i \in B(x,r)\}} \geq (1+\delta)nP(B(x,r))\right]$$

$$\leq \left(\frac{e^\delta}{(1+\delta)^{(1+\delta)}}\right)^{nP(B(x,r))}$$

$$\leq e^{-p_*(x)r^D n}\left(\frac{ep^* r^D n}{k}\right)^{kp_*(x)/p^*}$$

$\blacksquare$

The bound we prove below is written in a somewhat different form from the version of Lemma 4 in the main paper. This form follows somewhat more intuitively from Lemma 3, but does not make obvious the connection to the asymptotic Erlang distribution. To derive the form in the paper, one simply integrates the integral below by parts, plugs in the function $x \mapsto f\left(p_*(x)\Big/\frac{k/n}{c_D \varepsilon_k^D(x)}\right)$, and applies the bound $(e/k)^k \leq \frac{e}{\sqrt{k}\Gamma(k)}$.

**Lemma 4.** *Consider the setting of Lemma 3 and assume $\mathcal{X}$ is compact with diameter $\rho := \sup_{x,y \in \mathcal{X}} d(x,y)$. Suppose $f : (0,\rho) \to \mathbb{R}$ is continuously differentiable, with $f' > 0$. Then, for any $x \in \mathcal{X}$, we have the upper bound*

$$\mathbb{E}\left[f_+(\varepsilon_k(x))\right] \leq f_+\left(\left(\frac{k}{p_*(x)n}\right)^{\frac{1}{D}}\right) + \frac{(e/k)^k}{D(np_*(x))^{\frac{1}{D}}}\int_k^{np_*(x)\rho^D} e^{-y}y^{\frac{Dk+1-D}{D}}f'\left(\left(\frac{y}{np_*(x)}\right)^{\frac{1}{D}}\right)dy \tag{1}$$

*and the lower bound*

$$\mathbb{E}\left[f_-(\varepsilon_k(x))\right] \leq f_-\left(\left(\frac{k}{p^*n}\right)^{\frac{1}{D}}\right) + \frac{(e/\kappa(x))^{\kappa(x)}}{D\left(np_*(x)\right)^{\frac{1}{D}}}\int_0^{\kappa(x)} e^{-y}y^{\frac{D\kappa(x)+1-D}{D}}f'\left(\left(\frac{y}{np_*(x)}\right)^{\frac{1}{D}}\right)dy, \tag{2}$$

*where $f_+(x) = \max\{0, f(x)\}$ and $f_-(x) = -\min\{0, f(x)\}$ denote the positive and negative parts of $f$, respectively, and $\kappa(x) := kp_*(x)/p^*$.*

**Proof:** For notational simplicity, we prove the statement for $g(x) = f\left(np_*(x)x^D\right)$; the main result follows by substituting $f$ back in.

Define

$$\varepsilon_0^+ = f_+\left(\left(\frac{k}{p_*(x)n}\right)^{\frac{1}{D}}\right) \quad \text{and} \quad \varepsilon_0^- = f_-\left(\left(\frac{k}{p^*n}\right)^{\frac{1}{D}}\right).$$

Writing the expectation in terms of the survival function,

$$\mathbb{E}\left[f_+(\varepsilon_k(x))\right] = \int_0^\infty \mathbb{P}\left[f(\varepsilon_k(x)) > \varepsilon\right] d\varepsilon$$

$$= \int_0^{\varepsilon_0^+} \mathbb{P}\left[f(\varepsilon_k(x)) > \varepsilon\right] d\varepsilon + \int_{\varepsilon_0^+}^{f_+(\rho)} \mathbb{P}\left[f(\varepsilon_k(x)) > \varepsilon\right] d\varepsilon,$$

$$\leq \varepsilon_0^+ + \int_{\varepsilon_0^+}^{f_+(\rho)} \mathbb{P}\left[f(\varepsilon_k(x)) > \varepsilon\right] d\varepsilon, \tag{3}$$

since $f$ is non-decreasing and $\mathbb{P}\left[\varepsilon_k(x) > \rho\right] = 0$. By construction of $\varepsilon_0^+$, for all $\varepsilon > \varepsilon_0^+$, $f^{-1}(\varepsilon) > (k/(p_*(x)n))^{1/D}$. Hence, applying Lemma 3 followed by the change of variables

$y = np_*(x) \left(f^{-1}(\varepsilon)\right)^D$ gives [2]

$$\int_{\varepsilon_0^+}^{f_+(\rho)} \mathbb{P}\left[\varepsilon_k(x) > f^{-1}(\varepsilon)\right] d\varepsilon \leq \int_{\varepsilon_0^+}^{f_+(\rho)} e^{-np_*(x)\left(f^{-1}(\varepsilon)\right)^D} \left(\frac{enp_*(x)\left(f^{-1}(\varepsilon)\right)^D}{k}\right)^k d\varepsilon$$

$$= \frac{(e/k)^k}{D(np_*(x))^{\frac{1}{D}}} \int_k^{np_*(x)\rho^D} e^{-y} y^{\frac{kD+1-D}{D}} f'\left(\left(\frac{y}{np_*(x)}\right)^{\frac{1}{D}}\right) dy,$$

Together with (3), this gives the upper bound (1). Similar steps give

$$\mathbb{E}\left[f(\varepsilon_k(x))\right] \leq \varepsilon_0^- + \int_{\varepsilon_0^-}^{f_-(0)} \mathbb{P}\left[f(\varepsilon_k(x)) < -\varepsilon\right] d\varepsilon. \tag{4}$$

Applying Lemma 3 followed the change of variables $y = np_*(x)\left(f^{-1}(-\varepsilon)\right)^D$ gives

$$\int_{\varepsilon_0^-}^{f_-(\rho)} \mathbb{P}\left[\varepsilon_k(x) < f^{-1}(-\varepsilon)\right] d\varepsilon \leq \frac{(e/\kappa(x))^{\kappa(x)}}{D\left(np_*(x)\right)^{\frac{1}{D}}} \int_0^{\kappa(x)} e^{-y} y^{\frac{D\kappa(x)+1-D}{D}} f'\left(\left(\frac{y}{np_*(x)}\right)^{\frac{1}{D}}\right) dy$$

Together with inequality (4), this gives the result (2). $\blacksquare$

## B.1  Applications of Lemma 4

When $f(x) = \log(x)$, (1) gives

$$\mathbb{E}\left[\log_+(\varepsilon_k(x))\right] \leq \frac{1}{D}\log_+\left(\frac{k}{p_*(x)n}\right) + \left(\frac{e}{k}\right)^k \frac{\Gamma(k,k)}{D} \leq \frac{1}{D}\left(\log_+\left(\frac{k}{p_*(x)n}\right) + 1\right)$$

and (2) gives [3]

$$\mathbb{E}\left[\log_-(\varepsilon_k(x))\right] \leq \frac{1}{D}\left(\log_-\left(\frac{k}{p^*n}\right) + \left(\frac{e}{\kappa(x)}\right)^{\kappa(x)} \gamma(\kappa(x),\kappa(x))\right) \tag{5}$$

$$\leq \frac{1}{D}\left(\log_-\left(\frac{k}{p^*n}\right) + \frac{1}{\kappa(x)}\right). \tag{6}$$

For $\alpha > 0$, $f(x) = x^\alpha$, (1) gives

$$\mathbb{E}\left[\varepsilon_k^\alpha(x)\right] \leq \left(\frac{k}{p_*(x)n}\right)^{\frac{\alpha}{D}} + \left(\frac{e}{k}\right)^k \frac{\alpha\Gamma(k+\alpha/D,k)}{D(np_*(x))^{\alpha/D}}$$

$$\leq C_2 \left(\frac{k}{p_*(x)n}\right)^{\frac{\alpha}{D}}, \tag{7}$$

where $C_2 = 1 + \frac{\alpha}{D}$. For any $\alpha \in [-D\kappa(x), 0]$, when $f(x) = -x^\alpha$, (2) gives

$$\mathbb{E}\left[\varepsilon_k^\alpha(x)\right] \leq \left(\frac{k}{p^*n}\right)^{\frac{\alpha}{D}} + \left(\frac{e}{\kappa(x)}\right)^{\kappa(x)} \frac{\alpha\gamma(\kappa(x)+\alpha/D,\kappa(x))}{D(np_*(x))^{\alpha/D}} \tag{8}$$

$$\leq C_3 \left(\frac{k}{p^*n}\right)^{\frac{\alpha}{D}}, \tag{9}$$

where $C_3 = 1 + \frac{\alpha}{D\kappa(x)+\alpha}$.

# C  Proof of Bias Bound

**Theorem 5.** *Consider the setting of Lemma 3. Suppose Suppose $p$ is $\beta$-Hölder continuous, for some $\beta \in (0, 2]$. Let $f : (0, \infty) \to \mathbb{R}$ be differentiable, and define $M_f : \mathcal{X} \to [0, \infty)$ by*

$$M_f(x) := \sup_{z \in \left[\frac{p_*(x)}{\mu^*}, \frac{p^*}{\mu_*}\right]} \|\nabla f(z)\|$$

*(assuming this quantity is finite for almost all $x \in \mathcal{X}$). Suppose that*

$$C_M := \underset{X \sim p}{\mathbb{E}} \left[ \frac{M_f(X)}{(p_*(X))^{\frac{\beta}{D}}} \right] < \infty.$$

*Then, for $C_B := C_M L$,*

$$\left| \underset{X, X_1, \ldots, X_n \sim P}{\mathbb{E}} \left[ f(p_{\varepsilon_k(X)}(X)) \right] - F(p) \right| \leq C_B \left( \frac{k}{n} \right)^{\frac{\beta}{D}}.$$

**Proof:** By construction of $p_*$ and $p^*$,

$$p_*(x) \leq p_\varepsilon(x) = \frac{P(B(x, \varepsilon))}{\mu(B(x, \varepsilon))} \leq p^*.$$

Also, by the Lebesgue differentiation theorem [2], for $\mu$-almost all $x \in \mathcal{X}$,

$$p_*(x) \leq p(x) \leq p^*.$$

For all $x \in \mathcal{X}$, applying the mean value theorem followed by inequality (7),

$$
\begin{aligned}
\underset{X_1, \ldots, X_n \sim p}{\mathbb{E}} \left[ \left| f(p(x)) - f(p_{\varepsilon_k(x)}(x)) \right| \right] &\leq \underset{X_1, \ldots, X_n \sim p}{\mathbb{E}} \left[ \|\nabla f(\xi(x))\| \left| p(x) - p_{\varepsilon_k(x)}(x) \right| \right] \\
&\leq M_f(x) \underset{X_1, \ldots, X_n \sim p}{\mathbb{E}} \left[ \left| p(x) - p_{\varepsilon_k(x)}(x) \right| \right] \\
&\leq \frac{M_f(x) L D}{D + \beta} \underset{X_1, \ldots, X_n \sim P}{\mathbb{E}} \left[ \varepsilon_k^\beta(x) \right] \\
&\leq \frac{C_2 M_f(x) L D}{D + \beta} \left( \frac{k}{p_*(x) n} \right)^{\frac{\beta}{D}}
\end{aligned}
$$

Hence,

$$
\begin{aligned}
\left| \underset{X_1, \ldots, X_n \sim p}{\mathbb{E}} \left[ F(p) - \hat{F}(p) \right] \right| &= \left| \underset{X \sim p}{\mathbb{E}} \left[ \underset{X_1, \ldots, X_n \sim p}{\mathbb{E}} \left[ f(p(X)) - f(p_{\varepsilon_k(X)}(X)) \right] \right] \right| \\
&\leq \frac{C_2 L D}{D + \beta} \underset{X \sim p}{\mathbb{E}} \left[ \frac{M_f(X)}{(p_*(X))^{\frac{\beta}{D}}} \right] \left( \frac{k}{n} \right)^{\frac{\beta}{D}} = \frac{C_2 C_M L D}{D + \beta} \left( \frac{k}{n} \right)^{\frac{\beta}{D}}.
\end{aligned}
$$

∎

**Lemma 6.** *Let $c > 0$. Suppose there exist $b_\partial \in (0, \frac{1}{c})$, $c_\partial, \rho_\partial > 0$ such that for all $x \in \mathcal{X}$ with $\varepsilon(x) := \operatorname{dist}(x, \partial\mathcal{X}) < \rho_\partial$, $p(x) \geq c_\partial \varepsilon^{b_\partial}(x)$. Then,*

$$\int_{\mathcal{X}} (p_*(x))^{-c} \, d\mu(x) < \infty.$$

**Proof:** Let $\mathcal{X}_\partial := \{x \in \mathcal{X} : \operatorname{dist}(x, \partial\mathcal{X}) < \rho_\partial\}$ denote the region within $\rho_\partial$ of $\partial\mathcal{X}$. Since $p_*$ is continuous and strictly positive on the compact set $\mathcal{X} \backslash \mathcal{X}_\partial$, it has a positive lower bound $\ell := \inf_{x \in \mathcal{X} \backslash \mathcal{X}_\partial}$ on this set, and it suffices to show

$$\int_{\mathcal{X} \backslash \mathcal{X}_\partial} (p_*(x))^{-c} \, d\mu(x) < \infty.$$

For all $x \in \mathcal{X}_\partial$,

$$p_*(x) \geq \frac{\min\{\ell, c_\partial \varepsilon^{b_\partial}(x)\}}{\mu(B(x, \sqrt{D}))}.$$

Hence,

$$\int_{\mathcal{X} \backslash \mathcal{X}_\partial} (p_*(x))^{-c} \, d\mu(x) \leq \int_{\mathcal{X} \backslash \mathcal{X}_\partial} \ell^{-c} \, d\mu(x) + \int_{\mathcal{X} \backslash \mathcal{X}_\partial} c_\partial^{-c} \varepsilon^{-b_\partial/c}(x) \, d\mu(x).$$

The first integral is trivially bounded by $\ell^{-c}$. Since $\partial \mathcal{X}$ is the union of $2D$ "squares" of dimension $D - 1$, the second integral can be reduced to the sum of $2D$ integrals of dimension $1$, giving the bound

$$2D c_\partial^{-c} \int_0^{\rho_\partial} x^{-b_\partial/c}(x) \, dx.$$

Since $b_\partial/c < 1$, the integral is finite. ∎

For concreteness, we give an illustrative example of how Lemma 6 is useful.

**Example:** Consider the one-dimensional density $p(x) = (\alpha + 1)x^\alpha$ on $(0, 1)$. Though the lower bound $p_*$ provided by Lemma 2 is somewhat loose in this case, notice that, for $x < r \in (0, 1)$,

$$\frac{P(B(x, r))}{\mu(B(x, r))} \geq \frac{(x + r)^{\alpha+1}}{2r} \geq \frac{(x(1 + 1/\alpha))^{\alpha+1}}{2x/\alpha} = \frac{\alpha(1 + 1/\alpha)^{\alpha+1}}{2} x^\alpha,$$

and, for $r < x \in (0, 1)$,

$$\frac{P(B(x, r))}{\mu(B(x, r))} = \frac{(x + r)^{\alpha+1} - (x - r)^{\alpha+1}}{2r} \geq \frac{2r x^\alpha}{2r} = x^\alpha.$$

In either case, for $C_\alpha := \min\left\{1, \alpha(1 + 1/\alpha)^{\alpha+1}/2\right\}$, we have

$$p_*(x) := C_\alpha x^\sigma \leq \frac{P(B(x, r))}{\mu(B(x, r))}.$$

Thus, we have a local lower bound $p_*$ of the form in Lemma, satisfying the conditions of Lemma 6 with $b_\partial = \alpha$.

Now consider more general densities $p$ on $(0, 1)$. If $p(0) = 0$ and $p$ is right-differentiable at $0$ with $\lim_{h \to 0} \frac{p(h)}{h} > 0$ (i.e., the one-sided Taylor expansion of $p$ at $0$ has a non-zero first-order coefficient), then, near $0$, $p$ is proportional to $x$. This intuition can be formalized to show that the example above extends to quite general distributions.

## D  Proof of Variance Bound

**Theorem 7. (Variance Bound)** *Suppose that $\mathcal{B} \circ f$ is continuously differentiable and strictly monotone. Assume that $C_{f,p} := \mathbb{E}_{X \sim P}\left[\mathcal{B}^2(f(p_*(X)))\right] < \infty$, and that $C_f := \int_0^\infty e^{-y} y^k f(y) < \infty$. Then, for*

$$C_V := 2\left(1 + N_{k,D}\right)(3 + 4k)\left(C_{f,p} + C_f\right), \quad \text{we have} \quad \mathbb{V}\left[\hat{F}_\mathcal{B}(P)\right] \leq \frac{C_V}{n}.$$

**Proof:** For convenience, define

$$H_i := \mathcal{B}\left(f\left(\frac{k/n}{\mu\left(B(X_i, \varepsilon_k(X_i))\right)}\right)\right).$$

By the Efron-Stein inequality [1] and the fact that the $\hat{F}_\mathcal{B}(P)$ is symmetric in $X_1, \ldots, X_n$,

$$\mathbb{V}\left[\hat{F}_\mathcal{B}(P)\right] \leq \frac{n}{2} \mathbb{E}\left[\left(\hat{F}_\mathcal{B}(P) - F'_\mathcal{B}(P)\right)^2\right]$$

$$\leq n \mathbb{E}\left[\left(\hat{F}_\mathcal{B}(P) - F_{2:n}\right)^2 + \left(\hat{F}'_\mathcal{B}(P) - F_{2:n}\right)^2\right]$$

$$= 2n \mathbb{E}\left[\left(\hat{F}_\mathcal{B}(P) - F_{2:n}\right)^2\right],$$

where $\hat{F}'_{\mathcal{B}}(P)$ denotes the estimator after $X_1$ is resampled, and $F_{2:n} := \frac{1}{n}\sum_{i=2}^{n} H_i$. Then,

$$n(\hat{F}_n(P) - F_{2:n}) = H_1 + \sum_{i=2}^{n} 1_{E_i}(H_i - H'_i),$$

where $1_{E_i}$ is the indicator function of the event $E_i = \{\varepsilon_k(X_i) \neq \varepsilon'_k(X_i)\}$. By Cauchy-Schwarz followed by the definition of $N_{k,D}$,

$$n^2(\hat{F}_n(P) - \hat{F}_{n-1}(P))^2 = \left(1 + \sum_{i=2}^{n} 1_{E_i}\right)\left(H_1^2 + \sum_{i=2}^{n} 1_{E_i}(H_i - H'_i)^2\right)$$

$$= (1 + N_{k,D})\left(H_1^2 + \sum_{i=2}^{n} 1_{E_i}(H_i - H'_i)^2\right)$$

$$\leq (1 + N_{k,D})\left(H_1^2 + 2\sum_{i=2}^{n} 1_{E_i}\left(H_i^2 + H_i'^2\right)\right).$$

Taking expectations, since the terms in the summation are identically distributed, we need to bound

$$\mathbb{E}\left[H_1^2\right], \tag{10}$$

$$(n-1)\,\mathbb{E}\left[1_{E_2}H_2^2\right] \tag{11}$$

$$\text{and} \quad (n-1)\,\mathbb{E}\left[1_{E_2}H_2'^2\right]. \tag{12}$$

**Bounding** (10)**:** Note that

$$\mathbb{E}\left[H_1^2\right] = \mathbb{E}\left[\mathcal{B}^2\left(f\left(\hat{p}_k(X_1)\right)\right)\right] = \mathbb{E}\left[\mathcal{B}^2\left(g\left(\frac{p_*(x)}{\hat{p}_k(x)}\right)\right)\right]$$

for $g(y) = f(p_*(x)/y)$. Applying the upper bound in Lemma 4, if $\mathcal{B}^2 \circ g$ is increasing,

$$\mathbb{E}\left[H_1^2\right] \leq \mathcal{B}^2(g(1)) + \frac{e\sqrt{k}}{\Gamma(k+1)}C_\uparrow = \mathcal{B}^2(f(p_*(x))) + \frac{e\sqrt{k}}{\Gamma(k+1)}C_\uparrow.$$

If $\mathcal{B}^2 \circ g$ is decreasing, we instead use the lower bound in Lemma 4, giving a similar result. If $\mathcal{B}^2 \circ g$ is not monotone (i.e., if $\mathcal{B} \circ g$ takes both negative and positive values), then, since $\mathcal{B} \circ f$ *is* monotone (by assumption), we can apply the above steps to $(\mathcal{B} \circ g)_-$ and $(\mathcal{B} \circ g)_+$, which *are* monotone, and add the resulting bounds.

**Bounding** (11)**:** Since $\{\varepsilon_k(X_2) \neq \varepsilon'_k(X_2)\}$ is precisely the event that $X_1$ is amongst the $k$-NN of $X_2$, $\mathbb{P}\left[\varepsilon_k(X_i) \neq \varepsilon'_k(X_i)\right] = k/(n-1)$. Thus, since $E_2$ is independent of $\varepsilon_k(X_2)$ and

$$(n-1)\,\mathbb{E}\left[1_{E_2}H_2^2\right] = (n-1)\,\mathbb{E}\left[1_{E_2}\right]\mathbb{E}\left[H_2^2\right] = k\,\mathbb{E}\left[H_2^2\right] = k\,\mathbb{E}\left[H_1^2\right],$$

and we can use the bound for (10).

**Bounding** (12)**:** Since $E_2$ is independent of $\varepsilon_{k+1}(X_2)$ and

$$(n-1)\,\mathbb{E}\left[1_{E_2}H_2'^2\right] = (n-1)\,\mathbb{E}\left[1_{E_2}\mathcal{B}^2\left(f\left(\hat{p}_{k+1}(X_2)\right)\right)\right]$$

$$= (n-1)\,\mathbb{E}\left[1_{E_2}\right]\mathbb{E}\left[\mathcal{B}^2\left(f\left(\hat{p}_{k+1}(X_2)\right)\right)\right] = k\,\mathbb{E}\left[\mathcal{B}^2\left(f\left(\hat{p}_{k+1}(X_2)\right)\right)\right].$$

Hence, we can again use the same bound as for (10), except with $k+1$ instead of $k$.

Combining these three terms gives the final result.

∎

## Footnotes

[1]$B(x, r) := \{y \in \mathbb{X} : d(x, y) < r\}$ denotes the open ball of radius $r$ centered at $x$.

[2] $f$ need not be surjective, but the generalized inverse $f^{-1}: [-\infty, \infty] \to [0, \infty]$ defined by $f^{-1}(\varepsilon) := \inf\{x \in (0, \infty) : f(x) \geq \varepsilon\}$ suffices here.

[3] $\Gamma(s,x) := \int_x^\infty t^{s-1}e^{-t}\,dt$ and $\gamma(s,x) := \int_0^x t^{s-1}e^{-t}\,dt$ denote the upper and lower incomplete Gamma functions respectively. We used the bounds $\Gamma(s,x), x\gamma(s,x) \leq x^s e^{-x}$.