[Reviews · NeurIPS 2016]

Reviewer 1

Summary

The paper analyzes the performance of k-NN density functional estimators for fixed value of k. The paper considers Holder continuous densities with b \in [0,2]. The minimax rates for this problem was derived in Birge, and Massart, however most estimators are sub-optimal, with the exception of Krishnamurthy etal. However, their complexity is exponential in D, and cubic in n. In comparison, for fixed k, the k-NN estimators can be computed in time O(Dn^2). The main result of the paper is an upper bound on the convergence rate of fixed k-NN estimators.

Qualitative Assessment

The paper studies k-NN density functional estimators, and for Holder continuous densities show an upper bound on the convergence rate. This is a nice result. I am not convinced about two things. One is the emphasis on the smoothness assumptions on the boundary, and whether indeed it is a hinderance or a mere technicality. Also the authors can provide a few more details on the results of Baiu and Devroye. I found the following paper very related to the current submission. Shashank Singh Barnabas Poczos, "Analysis of k-Nearest Neighbor Distances with Application to Entropy Estimation", ICML 2016 I would encourage the authors to make a comparison with the paper. I am willing to upgrade my score on the originality/novelty if the authors can point the differences and improvements.

Confidence in this Review

1-Less confident (might not have understood significant parts)


Reviewer 2

Summary

This paper provides finite sample bounds for the performance of k-nearest neighbor estimators of functionals of probability densities. That is, the goal is to estimate a quantity of the form $F(P) = E[f(p(X))]$ when the probability density $p$ is unknown. The bounds assume the use of $k$-nearest neighbor density estimator, $pˆk(x)$, which allows for non-parametric estimates of continuous densities based on countable number of samples. The estimator $pˆk(x)$ is consistent if $k$ is allowed to go to infinity, and biased for a fixed $k$. One approach is to use a consistent sequence of estimators $pˆk(x)$ as a plug-in estimator, but it has the disadvantage that, for large $k$, this is difficult to compute. The approach taken in this work is to fix $k$ and correct for the bias. The work does not actually compute the bias correcting terms (these are well known for many functions of interest). Instead, it assumes that this bias correction is known and analyzes the performance of bias-corrected estimators of $F(P)$ (which are asymptotically unbiased).

Qualitative Assessment

This is a nice contribution to the literature on density functional estimation. The authors present a unified analysis of bias-corrected $k-NN estimators of functionals of probability densities and obtain finite-sample bounds on their bias and variance. The presentation is very clear, and all the assumptions are carefully stated. Minor comments: 1. Line 76: What is $c_D \epsilon^D_k(x)$? Did the authors mean $c_{D,r} \epsilon^D_k(x)$? 2. Line 174: Any function $p_*$ or a continuous function $p_*$? 3. Lines 230-231: The quantity $\hat{F}_{{\cal B}}(P)$ is a random quantity that depends on $X_1,\ldots,X_n$. Is there an expectation missing around $| \hat{F}_{{\cal B}}(P) - F(P)|$?

Confidence in this Review

2-Confident (read it all; understood it all reasonably well)


Reviewer 3

Summary

This paper addresses an important problem, of estimating functionals of a probability distribution, based on k-nearest neighbor statistics of independent samples. The results are all linked to a fixed-k setting.

Qualitative Assessment

There are some issues with the conditions imposed on distributions in this paper. In particular, regarding the local bound hypothesis: in the proof of Lemma 2 epsilon may depend on x, so p^* will be smaller than stated. As an example, if p(x) goes to 0 as x^a when x goes to 0, then p^*(x) goes to 0 as x^(a+1). This is problematic with the assumption that \int p_*(x)^(-\beta/d) dx < \infty. +++++++++++++++++ AFTER REBUTTAL: 1) The rebuttal here is helpful, but there seem to still be some issues with the local lower bounds hypothesis. They claim that it is very mild in light of Lemma 2, but Lemma 2 doesn't apply very well to the situation of p=x^a. I think that they should at least include what they've written in the rebuttal. The assumption for Shannon entropy that \int p_*(x)^(-beta/d) dx < infty is still a very strong assumption on how p may go to zero at the boundary of the support (which must have finite Lebesgue measure), and they do insist that p does go to zero at the boundary of the support. +++++++++++++++++ The results are globally worse than in [Berrett, Samworth, Yuan (16)], which is not cited. They have much less restrictive assumptions and Prop 3 in this paper provides a better bias bound. The assumption here requires, as an example that the support of the distribution has finite Lebesgue measure. The variance bound is also very crude. Indeed, the bound is used to suggest that fixed k is best, and is of order O(k^2 /n), whereas the correct order of the variance in O(1/n) in many cases. It is claimed that the variance bound is no larger than the minimax MSE rate which is incorrect. +++++++++++++++++ AFTER REBUTTAL: 2,3) It is true that increasing k won't improve the rate of convergence, but it can improve the constant factor significantly and to claim that fixed k is the best isn't necessarily true. Indeed, looking at Theorem 1 in the following paper: http://arxiv.org/pdf/1602.07440v1.pdf. There they find the asymptotic variance when k=1 and it is larger than the asymptotic variance when k diverges. It seems that from a statistical point of view,fixing k isn't the right thing to do, and that in practice larger k will be chosen for larger sample sizes. If k is chosen to diverge, their variance bound gives the wrong order, though it is the right order for fixed k.

Confidence in this Review

3-Expert (read the paper in detail, know the area, quite certain of my opinion)


Reviewer 4

Summary

The paper gives a finite sample analysis (bias and variance bounds) for a framework of nonparametric methods, estimating functionals of densities (such as divergences and entropies) by plugging in a k-nearest neighbor density estimate and taking a sample average. The interesting property of the methods being analyzed is that instead of increasing k with the sample size-- which leads to consistent density estimates-- k is fixed. The implication of considering fixed k is that the variance of the density estimate does not go to zero asymptotically. The non vanishing variance in the density estimate manifests as bias in the estimate of the functional. The methods being considered do some correction to obtain an asymptotically unbiased estimator. The variance of the functional estimate is asymptotically 0, in spite of the non vanishing variance of the density estimate, because of a sample averaging. The paper derives finite sample bounds for the bias and the variance of such methods under certain weak assumptions on the underlying density. The paper argues that the bounds are superior to most of the state of art approaches and those approaches that might have better convergence are computationally too demanding, concluding that the fixed-k approach with bias correction gives competitive convergence with significant computational advantage.

Qualitative Assessment

Congratulations to the authors for their excellent work. Quality: This is a high quality paper with significant theory and sophisticated math. The claims of the paper are well supported by the derived error bounds, though, I have a some concerns about the math at a couple of places (see under Concerns). The authors have clearly stated the assumptions under which the bounds apply and have given an objective comparison of their work with previously known results. Clarity: Overall the paper is very well written and reads easily. The authors have provided intuition which makes it easy to follow the math. The assumptions are very well motivated and discussed. Though there are some formatting issues, typos and notation reuse. Originality: The error bounds proved in the paper are novel. The proofs are non trivial and interesting. The authors have done an exceptional job of discussing the related work. Significance: The problem of estimating functionals of densities is very important in several machine learning and statistics problems. The competitive error bounds derived in the paper and the computational advantage of fixed-k nearest neighbor based methods justify using them for estimation. Main concern: As I understand, the argument used to apply lemma 4 to derive bounds on $E[f(\epsilon_k(x))]$, is that $f(\epsilon_k(x))$ can be expressed as $f(h(z))$ where $z=\frac{p_*(x)}{\hat{p}_k(x)}$ and h(z)=(\frac{kz}{cnp_*(x)})^{1/D}, and $f o h$ is an increasing function. However, the definition of $h$ is not independent of z; its definition changes with its argument $z$ because it contains $p_*(x)$ as well in the denominator. In this light, it is not straightforward how lemma 4 can be applied. The appendix version of lemma 4 seems to derive the bounds on $E[f(\epsilon_k(x))]$ directly. The proof starts with defining $g$, but never uses it. Also, it might be better to have the same result in the appendix and the main paper or lemma with a detailed proof should be added to derive one from the other. (Line 236) Expression for C_f when f(x)=log(x) is given. It seems like the M_{f,p}(x) is evaluated to be 1. However, I do not see how it can be a constant. Also the expectation should be with respect to p, but the integral is with respect to the borel measure. In lemma 4, line 218 says that the inequality 7 is a lower bound, however, it is actually an upper bound. Also upper bound for E(f) =E(f_+)-E(f_-) cannot be derived with <= in 7. From the expression for the variance bound (line 254), it seems that the upper bound increases with k, instead of decreasing as one would expect. Minor comments: There are some typos below Theorem 5: line 233 M_{f,o}, Equation below line 233 the second term should be E w.r.t q, not p. Line 232 gives the impression that q and p should have the same L, it might be better to have L_1, L_2. Before line 235, it would be good to have expression for the bias bounds for functionals of p and q and perhaps a lemma in the appendix. Notation reuse: r is used as the exponent in the norm as well as the radius of the ball. It would be good to include proofs that all the functionals in table 1 satisfy the assumptions of the theorems. The intuition provided in line 208 about asymmetry and bias is not is not obvious. Line 274 about testing problems is difficult to parse. Overall, I recommend accepting the paper if they address my concerns. After author's response: The authors have responded to my concerns satisfactorily. Though, I would like to suggest them to add few technical comments to Lemma 4 and Theorem 5 for more clarity.

Confidence in this Review

2-Confident (read it all; understood it all reasonably well)


Reviewer 5

Summary

This paper establishes the finite sample bounds for fixed-kNN density functionals estimator. The analysis applies for a family of estimators with known bias correction terms.

Qualitative Assessment

Overall I think the paper is well written, addresses an important question in density functional estimation. The math seems rigorous and the assumptions are motivated. Below are some concerns. 1. The author claims in line 87 the superior of their method over plug-in estimator because \hat{p} induces bias. My question is what if the bias correction of \hat{p} is known for fixed k? Do the two methods end up equivalent? If not, is the bound still better? 2. I would like to see more discussion about the relationship between the assumptions. For example, between A1 and A4 the author claim in line 172 that A4 is "much milder", but is it true that Lemma 2 shows A4 implies A1? Also between A2 and A3, it's not obvious to me which is stronger. Does A2 work for atom points, i.e., those who have non-zero probability mass? 3. Even for a theory paper, it's still of interest to show some simulations to illustrate the ideas, for example, the tightness of the upper bounds given in the paper, or the computational savings discussed here by using small k. Minor corrections: line 11: a number of; line 50: to; line 87: section 5

Confidence in this Review

2-Confident (read it all; understood it all reasonably well)


Reviewer 6

Summary

This paper analyzes the k-Nearest-Neighbor based estimator proposed for general functionals of densities. The authors derive upper bounds on the bias and variance of the estimator. The approach is based on deriving bounds on the moments of KNN distances. The benefit of this estimator compared to the previous KNN-based estimators is that the estimator is asymptotically unbiased and consistent for fixed k. This provides some of the first results on the bias and variance of several popular estimators when the bias at the boundary is taken into consideration. The essential assumption for proposing such estimators is knowledge about the bias correction function, which creates some complexities for practical use.

Qualitative Assessment

This paper provides a new way of analyzing k-nn based entropy and divergence estimators. In order to analyze the convergence rates, most papers assume that the densities are bounded away from zero. The authors of this paper provide an alternative approach where the densities are locally lower bounded. This approach may be useful in the analysis of other estimators of information measures. However, it is not exactly clear if the assumptions are practical. See below for more details. 1- The paper could benefit from a little more proofreading. For example, the final sentence on line 273 does not make sense grammatically. 2- The paper didn’t provide any numerical simulations to validate the derived rates for convergence. While this is not required for publication, it would strengthen the paper to show simulations that compare the convergence of the estimator to the theoretical bounds. Also it would be nice if authors could compare the convergence rates for different values of k, by numerical experiments. 3- The examples given as bias correction functions in Table 1 are generally for asymptotic settings. For example, the multiplicative constants for Renyi alpha entropy and divergence derived in “Leonenko and Pronzato [2010]”, and “Poczos and Schneider [2011]” are such the estimator is asymptotically unbiased and consistent. But for finite-sample case, we cannot make sure that with these choices of k, the fact that with these choices of k, relation (3) holds for every n, is not clear. The authors mention that finding the bias correction function is not addressed in this paper, and they assume it given, however, they should at least show that such function exists the functions in Table 1. Basically, the question is that under what conditions we can make sure that some bias correction function exists such that it makes the relation (3) true for any choice of n. 4- The authors mention that they replace assumption (A1) with a much milder assumption that p is locally lower bounded on its support as discussed in the paper. In conjunction with assumption A2, this assumption appears to be more restrictive than A1. It would be good for the authors to provide some examples of distributions that fulfill A4 and A2 to verify that this approach is practical. 5- A common method for dealing with the bias at the boundary is to assume continuity at the boundary by extending the density beyond the boundary (e.g. Sricharan et al (2013)). Can the authors comment briefly on a comparison between their method and the extension approach? Is the authors' approach easier? 6- In the bias bound and variance bounds, it is assumed that C_f <\inf and C_{f,p}<\inf, but what do these conditions mean in terms of the distribution? The authors attempt to explain this by providing a sufficient condition in Lemma 6. However, the conditions in the lemma are not intuitive. Please provide more details. An example would be helpful.

Confidence in this Review

2-Confident (read it all; understood it all reasonably well)